# Masked Diffusion Models Are Fast Distribution Learners

## Abstract

Diffusion models have emerged as the *de-facto* generative model for image synthesis, yet they entail significant training overhead, hindering the technique's broader adoption in the research community. We observe that these models are commonly trained to learn all fine-grained visual information from scratch, thus motivating our investigation on its necessity. In this work, we show that it suffices to set up pre-training stage to initialize a diffusion model by encouraging it to learn some primer distribution of the unknown real image distribution. Then the pre-trained model can be fine-tuned for specific generation tasks efficiently. To approximate the primer distribution, our approach centers on masking a high proportion (e.g., up to 90%) of an input image and employing masked denoising score matching to denoise visible areas. Utilizing the learned primer distribution in subsequent fine-tuning, we efficiently train a ViT-based diffusion model on CelebA-HQ $256 \times 256$ in the raw pixel space, achieving superior training acceleration compared to denoising diffusion probabilistic model (DDPM) counterpart and a new FID score record of 6.73 for ViT-based diffusion models. Moreover, our masked pre-training technique can be universally applied to various diffusion models that directly generate images in the pixel space, aiding in the learning of pre-trained models with superior generalizability. For instance, a diffusion model pre-trained on VGGFace2 attains a 46% quality improvement through fine-tuning on only 10% data from a different dataset. Our code will be made publicly available.

## 1 Introduction

Diffusion models or score-based generative models (Ho et al., 2020; Sohl-Dickstein et al., 2015; Song et al., 2020b; Song & Ermon, 2019) have demonstrated exceptional performance in image synthesis and emerged as the state-of-the-art learning models for this task. The core denoising training approach has also been quickly adopted in various tasks such as image editing (Saharia et al., 2022a; Goel et al., 2023; Avrahami et al., 2022; Singh et al., 2023) and controllable image generation (Ramesh et al., 2021; 2022; Saharia et al., 2022b; Epstein et al., 2023; Qiu et al., 2023; Ruiz et al., 2023; Chen et al., 2023; Balaji et al., 2022). However, this approach is commonly adopted by training a model is to simultaneously learn all fine-grained visual details presented in images throughout the denoising training process, demanding intensive computational resources, especially for generating high-resolution images. In this work, we try to investigate if denoising training can benefit from avoiding modeling the complete raw image data in the early training stage and approach enhancing the overall training efficiency from a perspective that can be taken in tandem with previous studies.

We start by describing our approach by take painting as an intuitive example. Rather than directly traversing all fine-grained details, a painter usually starts with more distinguishing features, such as the global structure or local prominent texture. For training diffusion models, we anticipate that this natural decomposition can also be applied in analogy, making training comparably easier by first approximating some "primer" distributions that preserve principle or salient features of target images. The model can also be relieved from inspecting all intricate details, which contributes significantly to training difficulty. As such, the subsequent modeling of detailed image information can be effectively accelerated. However, it is non-trivial to learn such primer distributions from the real distribution which is unknown by itself. It can be resource-intensive to be collect accurate supervising signals or annotations if we follow the canonical supervised manner. To address this challenge, we first define a *primer* distribution as one that shares the same group of marginals,

which contains diverse important features, with the target data distribution. There exist many primer distributions that satisfy this condition while each can be regarded as a specific instance that can be further transformed into the target distribution. Then we propose a simple yet effective approach to implicitly approximate this primer distribution by modeling all marginals of the target distribution. Specifically, we apply random masking to every image input to a diffusion model. We also incorporate the positional information of the visible pixels into the model input as additional clues to distinguish different marginal distributions. Each masked input image can be regarded as a sample drawn from some arbitrary marginal distribution. We consider this approach as sharing a close spirit as Dropout Srivastava et al. (2014), which essentially learns a distribution of models. By performing denoising training on the visible parts, we try to approximately learn a joint distribution that composed of various marginal distributions, which can be aggregated to preserve meaningful local or global feature patterns. As such, this step enables a preferable initialization point for modeling the target distribution.

Consequently, the prevalent end-to-end process for training a diffusion model can be decomposed into a two-stage path: the first *masked pre-training* stage, which masks parts of the input image and performs masked denoising score matching (MDSM) on visible parts, followed by *denoising fine-tuning* equipped with the conventional weighted denoising score matching (DSM) objective (Ho et al., 2020; Vincent, 2011) as the second stage. The mask rate or number of marginalized variables, and the mask sampling strategy can be chosen empirically and remain fixed throughout the training process (we used multiple mask rates in our major experiments). It is important to note that, MDSM can be adopted as a plug-and-play technique and integrated with existing diffusion models. The generalizability obtained by the pre-trained model facilitate the subsequent stage by accelerating the training process across various datasets and training paradigms, as well as improving training diffusion models on limited data. Therefore, in the subsequent stage, the model endeavors to capture more informative details prevalent in the training data. We name the models yielded by our training framework as **Masked Diffusion Models** (MaskDM).

The **contributions** of our work can be summarized as follows:

(i) We design a two-stage training framework for improving diffusion model training efficiency. We propose to incorporate different masking strategies into the pre-training stage to build a pre-trained model that has desirable generalizability to facilitate efficiency fine-tuning for different downstream image synthesis tasks. We conduct thorough experiments to investigate different masking configurations for their impacts on both model performance and efficiency improvement, and provide practical guidance for applying our proposed framework.

(ii) We design our pre-training stage to be compatible with arbitrary diffusion-based algorithms and apply our proposed framework to reduce the substantial computational burden for successfully training ViT-based diffusion models on the high-resolution CelebA-HQ $256 \times 256$ dataset in the pixel space on 2 A100 GPU cards.

(iii) We conduct experiments to train ViT-based diffusion models on different image synthesis datasets and achieve superior cost reduction for training a DDPM (Ho et al., 2020) counterpart while obtaining better image generation performance. The experiment results also demonstrate excellent generalizability of pre-train models, which allows fine-tuning with only 10% of training data to achieve an improvement of 46% in the performance for the downstream tasks.

## 2 PRELIMINARY ON DIFFUSION MODELS

The diffusion model (Sohl-Dickstein et al., 2015; Ho et al., 2020) is composed of a forward and a reverse process. The forward process is defined as a discrete Markov chain of length $T$: $q(\boldsymbol{x}_{1:T}|\boldsymbol{x}_0) = \prod_{t=1}^{T} q(\boldsymbol{x}_t|\boldsymbol{x}_{t-1})$. For each step $t \in [1, T]$ in the forward process, a diffusion model adds noise $\epsilon_t$ sampled from the Gaussian distribution $\mathcal{N}(0, \mathbf{I})$ to data $\boldsymbol{x}_{t-1}$ and obtains disturbed data $\boldsymbol{x}_t$ from $q(\boldsymbol{x}_t|\boldsymbol{x}_{t-1}) = \mathcal{N}(\boldsymbol{x}_t; \sqrt{1-\beta_t}\boldsymbol{x}_{t-1}, \beta_t^2\mathbf{I})$. $\beta$ determines the scale of added noise at each step and can be prescribed in different ways (Ho et al., 2020; Nichol & Dhariwal, 2021) such that $p(\boldsymbol{x}_T) \approx \mathcal{N}(0, \mathbf{I})$. Noticeably, instead of sampling sequentially along the Markov chain, we can sample $x_t$ at any time step $t$ in the closed form via $q(\boldsymbol{x}_t|\boldsymbol{x}_0) = \mathcal{N}(\boldsymbol{x}_t; \sqrt{\bar{\alpha}_t}\boldsymbol{x}_0, (1-\bar{\alpha}_t)I)$, where $\bar{\alpha}_t = \prod_{s=1}^{t}(1-\beta_s)$. The reverse process is also defined as a Markov chain: $p_\theta(\boldsymbol{x}_{0:T}) = p(\boldsymbol{x}_T)\prod_{t=1}^{T} p_\theta(\boldsymbol{x}_{t-1}|\boldsymbol{x}_t)$. In DDPM Ho et al. (2020), $p_\theta(\boldsymbol{x}_{t-1}|\boldsymbol{x}_t)$ is parameterized as $\mathcal{N}(\boldsymbol{x}_t; \mu_\theta(\boldsymbol{x}_t, t), \sigma_t)$, where $\mu_\theta(\boldsymbol{x}_t, t) =$

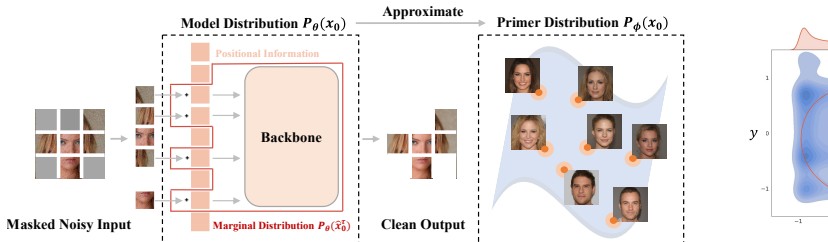

Figure 1: Illustration of the pre-training stage.  Figure 2: 2D Swiss roll example.

$\frac{1}{\sqrt{\alpha_t}}(\boldsymbol{x}_t - \frac{\beta_t}{\sqrt{1-\bar{\alpha}_t}}\boldsymbol{\epsilon}_\theta(\boldsymbol{x}_t, t))$ and $\sigma_t$ is a time-dependent constant. Given $\boldsymbol{x}_t$ and the time step $t$, $\boldsymbol{\epsilon}_\theta$ is a neural network and aims at predicting the noise $\boldsymbol{\epsilon} \sim \mathcal{N}(0, \mathbf{I})$ used to construct $\boldsymbol{x}_t$ together with $\boldsymbol{x}_{t-1}$. Using this parameterization, the variational objective in Sohl-Dickstein et al. (2015) is ultimately simplified to Eq.1, which can be seen as a variant of DSM Vincent (2011) over multiple noise scales.

$$L_{simple}(\theta) = \mathbb{E}_{t,\boldsymbol{x}_0,\boldsymbol{\epsilon}}\left[\left\|\boldsymbol{\epsilon} - \boldsymbol{\epsilon}_{\boldsymbol{\theta}}(\sqrt{\bar{\alpha}_t}\boldsymbol{x}_0 + \sqrt{1-\bar{\alpha}_t}\boldsymbol{\epsilon}, t)\right\|^2\right]. \tag{1}$$

In our two-stage training framework, we mainly adopt the vanilla training procedure of DDPM explained above in the second denoising fine-tuning stage. Our training framework is also compatible with SDE Song et al. (2020b), the continuous variant of DDPM (investigated in Sec.4.4). Before the fine-tuning starts, the model is loaded with weights obtained via masked pre-training, which we introduced in Sec.3.1.

## 3 MASKED DIFFUSION MODELS

We present an intuitive explanation for our inspiration in Fig. 2. Assuming that we are approximating a 2D Swiss roll distribution $p(\boldsymbol{z})$ (represented by the red line), where $\boldsymbol{z} = (x, y)$. Fig.2 displays another distribution $p_\phi(\boldsymbol{z})$ with a blue heatmap, which fully covers the target distribution $p(\boldsymbol{z})$, traversing all modes of the Swiss roll distribution. In comparison with approximating $p(\boldsymbol{z})$ from scratch, gradually shaping a distribution that is initialized as $p_\phi(\boldsymbol{z})$ , which shares with $p(\boldsymbol{z})$ the same marginal distribution, i.e., $p(x)$ and $p(y)$, is expected to be comparably easier. A close analogy can be drawn between this intuition and initializing a neural network with different initialization techniques. In particular, for image data, as the dimensionality increases, the data space expands significantly faster than the space expanded by real image samples. As such, initializing a task for approximating a high-dimensional distribution $p(\boldsymbol{z})$ with $p_\phi(\boldsymbol{z})$ , which partially preserves the sophisticated relations between different marginal distributions, may bring even more computational benefits.

### 3.1 MASKED PRE-TRAINING

Following the aforementioned intuitive example, we denote an image $\boldsymbol{x_0}$[1] by a vector: $(x_0^1, x_0^2, , x_0^3, , ..., x_0^N)$ , where $N$ represents the number of pixels. Then the data distribution $p(\boldsymbol{x_0})$ can be expressed as the joint distribution of $N$ pixels. Let $\tau$ represents a randomly selected subsequence of $[1, ..., N]$ with a length of $S$. We denote the subset of selected pixels as $\{x_0^{\tau_i}\}_{i=1}^S$ and the resulting marginal distribution as $p(\hat{\boldsymbol{x}}_0^{\boldsymbol{\tau}}) = p(x_0^{\tau_1}, x_0^{\tau_2}, x_0^{\tau_3}, ..., x_0^{\tau_S})$. For simplicity, with $S$ being fixed, we utilize $\hat{\boldsymbol{x}}_0$ to represent any marginal variable combinations $\{\tau \in [1, ..., N], |\tau| = S \mid \hat{\boldsymbol{x}}_0^\tau\}$, and $p(\hat{\boldsymbol{x}}_0)$ to represent the corresponding marginal distribution. Then it is evident $p(\boldsymbol{x_0})$ belongs to a family $\mathcal{Q}$ of distributions that share the same set of marginal distributions $p(\hat{\boldsymbol{x}}_0)$. We introduce the term *primer* distribution to refer to any distribution in $\mathcal{Q}$ other than $p(\boldsymbol{x_0})$ that satisfies this condition. We represent such distributions using the notation $p_\phi(\boldsymbol{x_0})$, where $\phi$ represents the unknown true parameters of the distribution. It is non-trivial to approximate $p_\phi(\boldsymbol{x_0})$, particularly when the samples from $p_\phi(\boldsymbol{x_0})$ are not available. We initialize the task of approximating $p_\phi(\boldsymbol{x_0})$ with a diffusion model $p_\theta(\boldsymbol{x_0})$, defined as introduced in Sec.2. In each training iteration, by training with a batch of images sampled from some arbitrary marginal distributions, which can be further viewed as sampled from $p_\theta(\boldsymbol{x_0})$, we are implicitly approximating $p_\phi(\boldsymbol{x_0})$ by modeling all its marginals. This approach can be viewed as sharing a similar spirit as Dropout Srivastava et al. (2014), which optimizes different randomly constructed "thin" sub-networks to approximate a distribution of true models. To achieve

this, we mask each image input $\boldsymbol{x_0}$ with a vector $\mathbf{M} \in \{0,1\}^N$, and incorporate the positional information $\mathbf{H} \in R^N$ of the visible pixels into the model input as additional clues to distinguish different marginal distributions. In practice, we observe that this simple masking approach suffices to preserve meaningful visual details while enabling a much faster pre-training convergence, which further facilitates subsequent fine-tuning, hence reducing the overall training time.

The masking method is presented in Algorithm 1, where the variables to be masked are selected in an uniform and pixel-wise manner. The masked image $\hat{\boldsymbol{x}}_0$ and noise $\hat{\boldsymbol{\epsilon}}$ are then integrated to construct $\hat{\boldsymbol{x}}_t$ such that $\hat{\boldsymbol{x}}_t = \sqrt{\bar{\alpha}_t}\hat{\boldsymbol{x}}_0 + \sqrt{1-\bar{\alpha}_t}\hat{\boldsymbol{\epsilon}}$, following the forward process of diffusion model. Subsequently, we optimize the model parameters by following the below MDSM objective, a variant of the objective defined in Eq.1:

$$L_{mdsm}(\theta) = \mathbb{E}_{t,\hat{\boldsymbol{x}}_0,\hat{\boldsymbol{\epsilon}}}\left[\left\|\hat{\boldsymbol{\epsilon}} - \boldsymbol{\epsilon_\theta}(\sqrt{\bar{\alpha}_t}\hat{\boldsymbol{x}}_0 + \sqrt{1-\bar{\alpha}_t}\hat{\boldsymbol{\epsilon}}, t)\right\|^2\right]. \tag{2}$$

An overview of the proposed pipeline is illustrated in Fig.1. An example use case is shown as masking a face image with a set of grey square blocks. The masked image can be seen as a sample drawn from a marginal distribution that is identified by the selected square blocks, which marginalize out all covered pixels. Considering the positional information $H$ as some fixed or learnable parameters of the model, then $p_\theta(\boldsymbol{x_0})$ is also "marginalized" by applying masking to subsample $H$. As such, given sufficient training time, $p_\theta(\boldsymbol{x_0})$ converges to a certain primer distribution $p_\phi(\boldsymbol{x_0})$ from $\mathcal{Q}$, based on which we further approximate the true data distribution $p(\boldsymbol{x_0})$ via fine-grained denoising training. The details are discussed in Sec.2. Additionally, following the conventional sampling procedure (Ho et al., 2020; Song et al., 2020a) of diffusion models, we could draw samples from $p_\theta(\boldsymbol{x_0})$ or its marginal distributions by customizing $\mathbf{M}$.

## 3.2 MODEL ARCHITECTURE AND MASKING CONFIGURATION

In our implementation, we utilize U-ViT, a ViT-based backbone proposed in Bao et al. (2022), for the simplicity of applying different masking strategies. A predominant challenge in training ViT-based diffusion models is the substantial computational burden, which includes high CUDA memory usage and lengthy training times. We demonstrate in Fig.4 that our masked pre-training significantly boosts the training efficiency of ViT-based diffusion models on raw image data.

In practice, it is crucial to carefully configure the masking setting, including both $S$ (or the mask rate $m = 1 - \frac{S}{N}$) and the strategy for sampling the mask vector $\mathbf{M}$. Specifically, the mask rate $m$ determines the average degree of similarity between the true data distribution and the primer distributions such that a lower value of $m$ indicates a greater resemblance. Besides, given U-ViT as the back-bone, a mask is sampled as a group of neighbouring pixels instead of individual and independent pixels. As such, the sampled masks essentially determine the range of primer distributions that could be possibly learned. As illustrations, Fig.3b and Fig.3c display various samples from two different primer distributions, which are implicitly learned via different mask sampling strategies.

---

**Algorithm 1:** Masking method

**Input:** $\boldsymbol{x_0}$, $\boldsymbol{\epsilon}$, $\mathbf{H}$ and $S$
**Output:** $\hat{\boldsymbol{x}}_0$, $\hat{\boldsymbol{\epsilon}}$
$\mathbf{M} \leftarrow \mathbf{0}$;
$\mathbf{K} \leftarrow [1, ..., N]$;
**for** $i \leftarrow 1$ *to* $S$ **do**
    $j \sim U([1, ..., |\mathbf{K}|])$;
    Set $\mathbf{M}[j] \leftarrow 1$;
    Remove $\mathbf{K}[j]$ from $\mathbf{K}$;
$\hat{\boldsymbol{x}}_0 \leftarrow \mathbf{M} \odot (\boldsymbol{x_0} + \mathbf{H})$;
$\hat{\boldsymbol{\epsilon}} \leftarrow \mathbf{M} \odot (\boldsymbol{\epsilon} + \mathbf{H})$;
**return** $\hat{\boldsymbol{x}}_0$, $\hat{\boldsymbol{\epsilon}}$;

---

In this work, we have designed three different masking strategies, namely, patch-wise masking, block-wise masking, and cropping. Examples for each masking type are shown in Fig. 3a. Patch-wise masking entails the random occlusion of a predefined number of image patches. Block-wise masking involves randomly selecting image blocks for masking, where each block comprises a fixed quantity of image patches. Lastly, cropping entails randomly selecting a top-left coordinate and the corresponding fixed-size square region then masking the area outside the chosen square. We explore and compare a range of masking configurations in Sec.4.2

---

[0]We follow the conventions and denote a clean image as $\boldsymbol{x_0}$, where the subscript 0 is the time step.

## 4 EXPERIMENTS

### 4.1 EXPERIMENTAL SETUP

**Implementation details.** We compare with existing methods on two datasets: CelebA Liu et al. (2015), CelebA-HQ Karras et al. (2017). We employ our MaskDM models with the U-ViT model architectures introduced in Bao et al. (2022) with certain modifications. Specifically, we utilize the U-ViT-Small setup from Bao et al. (2022) as our MaskDM-S models and construct our MaskDM-B model by removing five transformer blocks from U-ViT-Mid to fit in 1 Tesla V100 GPU card given a single $256 \times 256$ image as input with a $4 \times 4$ patch size. In all MaskDMs, we discard the appending convolutional block initially appearing in the U-ViT model and find the performance to be trivially affected. Similar to the settings in (Bao et al., 2022), we conduct all experiments with mixed precision considering training efficiency and employ the AdamW optimizer with coefficients set to (0.99, 0.99). The maximum diffusion step $T$ is set to 1000. We maintain an exponential moving average (EMA) model during training and use the EMA model during sampling. Unless specified, we adopt 50% $2 \times 2$ block-wise masking when pre-training on $64 \times 64$ images, and adopt $4 \times 4$ block-wise masking on images of resolution no less than $128 \times 128$. Further detailed information is provided in the Appendix B.

**Evaluation Settings.** During the evaluation, we utilize Fréchet Inception Distance (FID) Heusel et al. (2017) to measure the quality of generated images. We mainly employ two different samplers, namely, Euler-Maruyama SDE sampler Song et al. (2020b) and DDIM Song et al. (2020a), to generate samples. When comparing with current methods, we compute FID scores on 50k generated samples, and we apply Euler-Maruyama SDE sampler with 1k sampling steps on CelebA $64 \times 64$ and DDIM sampler with 500 sampling steps on CelebA $128 \times 128$ and CelebA-HQ $256 \times 256$.

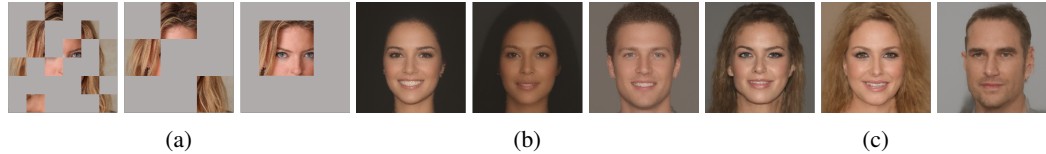

| (a) | (b) | (c) |

Figure 3: (a) Three masking strategies are examined in our experiments. From left to right, the strategies are represented as patch-wise masking, block-wise masking, and cropping. (b) and (c) Samples from primer distribution, captured utilizing patch-wise and block-wise masking respectively, given a mask rate of 90%. Notably, the model pre-trained with cropping at 90% mask rate exhibits limited capability in generating plausible samples; therefore, we do not illustrate the results here (See Appendix 8).

### 4.2 INVESTIGATING MASK CONFIGURATIONS

Table 1: Mask configuration investigation on CelebA $64 \times 64$, where pre-trained weights are acquired from different masking configurations. The baseline model, trained from scratch without loading pre-trained weights, is marked in gray.

(a) Impact of mask configuration

| Mask | 10% | 50% | 90% |
|---|---|---|---|
| patch | 6.85 | 6.58 | 7.34 |
| 2x2 block | **6.77** | **6.51** | 8.99 |
| 4x4 block | 6.92 | 6.88 | **6.91** |
| cropping | 6.92 | 6.82 | 8.62 |
| from scratch | | 7.55 | |

(b) Impact of computational budget

| Mask | Rate | Steps | bs=128 | bs=256 |
|---|---|---|---|---|
| patch | 10% | 50k | 6.85 | 6.31 |
| 2x2 block | 10% | 50k | 6.77 | 6.71 |
| 2x2 block | 50% | 50k | - | 6.51 |
| 2x2 block | 50% | 100k | - | 6.27 |
| 2x2 block | 50% | 150k | - | 6.05 |

(c) Impact of block size

| Mask | FID↓ |
|---|---|
| patch | 6.58 |
| 2x2 block | 6.51 |
| 4x4 block | 6.88 |
| 8x8 block | 7.43 |

To investigate the impact of masking strategy, we experiment with different types of masking, including patch-wise masking, block-wise masking, and cropping, to adjust masking granularity. Our masking strategies are demonstrated in Fig. 3a. Specifically, we pre-train models with different configurations on CelebA $64 \times 64$, using mask rates of 10%, 50%, and 90%, respectively. During pre-training, the GPU memory usage is fixed across different experiments and the default pre-training iterations are set as 50k in all experiments, for which we observe the pre-training curves are saturated.

Subsequently, given a pre-trained model, we fine-tune it for 200k steps to optimize the objective delineated in Eq. 1. To demonstrate the effect of adopting masked pre-training, we setup a baseline model which is trained on the same set of data from scratch (as detailed in Sec. 2) for 250k steps. This ensures comparable total training costs between the baseline model and its pre-trained counterpart.

**Comparing different masking types and mask rates.** As shown in Tab. 1(a), we first observe that models trained with cropping generally obtain the worst FID scores as we vary the mask rate. A possible explanation is that randomly cropped images retain limited global structural information, which constraints the model from building long-range connections among different variables. As a result, given a pair of fixed batch size and training step, cropping makes it more challenging for the diffusion model to capture the consistent critical visual features. On the other hand, block-wise masking (including both 2x2 and 4x4 block) achieves the best results across all settings, while patch-wise masking achieves the second best FID scores.

Moreover, by comparing the FID scores obtained by selecting different mask rates, we observe that pre-trained models paired with the 50% mask rate outperform other ratios in most cases. In particular, the model pre-trained with 50% 2x2 block-wise masking achieve an FID score of 6.51, which is significantly better than the baseline. We also notice that the models pre-trained with a 90% mask rate exhibit a rapid divergence in FID scores after 50k training steps. We delve deeper into the causes of this problem by studying different impact factors (detailed findings are presented in Appendix C) and find that the adopted linear noise schedule contributes significantly to training instability. This can be effectively mitigated by utilizing the cosine noise schedule (Nichol & Dhariwal, 2021).

In addition, the results presented in Tab. 1(a) show that different block sizes also impact the generation performance. By taking the patch-wise masking as a 1x1 block-wise masking, we explore this impact and demonstrate the results in Tab. 1(c). We observe that using masking with a larger block size generally leads to performance degradation. Therefore, in the following, we mainly focus on evaluating patch-wise and block-wise masking approaches with different mask rates.

**Delving into mask rate, batch size, and training steps.** In our experiments, we are particularly interested in the case where the overall computation resource is limited, which is common in academic research. More concretely, we assume there is a fixed GPU resource budget. Given this resource constraint, we confront a trade-off between mask rate and batch size. For instance, to maintain a constant GPU usage, when applying a lower mask rate, which consumes more CUDA memory per image, we are limited to pre-train a model with a smaller batch of data. Indeed, in our experiments, models using a mask rate of 10% consume $1.5\times$ more GPUs than those using a mask rate of 50%. This raises the question that the less competitive performance obtained by block-wise and patch-wise masking with a rate of 10% may result from an inadequate batch size of 128. To investigate this question, we enlarge the batch size for both aforementioned settings to 256. This setting corresponds to the case where the computation resources are sufficient to support larger batch size pre-training. As presented in Tab. 1(b), both models trained with block-wise and patch-wise masking with a rate of 10% and a batch size of 256 exhibit improved performance as expected.

We also find that a higher mask rate often requires fewer computing resources (i.e., GPUs) but slightly more training steps to achieve performance comparable to its lower mask rate counterparts. As such, we return to the case where the GPU memory capacity constraint still holds and confine to the previous best setting with a 2x2 block-wise masking and a mask rate of 50%. We continue upon the above investigation to employing a batch size of 256, exploring the effect of extending the resource constraint in terms of pre-training steps. Specifically, we increase the number of pre-training steps from 50k to 100k and 150k, respectively, and present the results in Tab. 1(b). We observe a clear trend of performance improvement as the number of pre-training steps increase. The results are in alignment with our expectation and indicate that a longer pre-training time is generally helpful for improving the overall training performance.

The above investigation indicates the importance of properly configuring the mask rate, batch size, and training step, for optimizing model performance while aligning with affordable computing resources. These empirical findings open the opportunity for designing an automated dynamic training schedule, similar to Successive Halving Jamieson & Talwalkar (2016), that balances the trade-off between these intertwined hyper-parameters under a constant training budget. In fact, we have explored manually adjusting the training schedule and obtained the best generation performance in Sec. 4.3. We leave a more systematic study of training schedule automation to our future work.

Table 2: FID results on CelebA $64 \times 64$

| Method | FID ↓ | Params |
|---|---|---|
| DDIM Song et al. (2020a) | 3.26 | 79M |
| U-ViT-small Bao et al. (2022) | 2.87 | 44M |
| PNDM Liu et al. (2022) | 2.71 | 79M |
| **MaskDM-S** | **2.27** | 44M |

Table 3: FID results on CelebA $128 \times 128$

| Method | FID ↓ | Params |
|---|---|---|
| Gen-ViT[†] Yang et al. (2022) | 22.07 | 12.9M |
| Baseline | 12.96 | 102M |
| **MaskDM-B** | **6.83** | 102M |

Table 4: FID results on CelebA-HQ $256 \times 256$. Results of latent diffusion models are listed in gray color

| Method | FID ↓ | Params |
|---|---|---|
| VQ-GAN Esser et al. (2021) | 10.2 | 355M |
| PGGAN Karras et al. (2017) | 8.03 | - |
| DDGAN Xiao et al. (2021) | 7.64 | - |
| LSGM Vahdat et al. (2021) | 7.22 | - |
| LDM-4 Rombach et al. (2022) | 5.11 | 274M |
| VESDE Song et al. (2020b) | 7.23 | 66M |
| Soft Truncation Kim et al. (2021) | 7.16 | 66M |
| P2 Weighting Choi et al. (2022) | 6.91 | 94M |
| **MaskDM-B** | **6.27** | 102M |

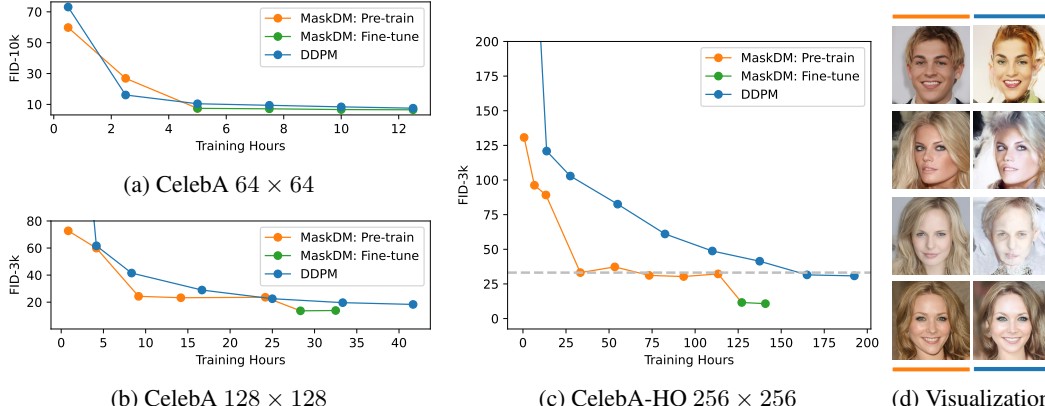

(a) CelebA $64 \times 64$

(b) CelebA $128 \times 128$

(c) CelebA-HQ $256 \times 256$

(d) Visualization

Figure 4: Comparison of training efficiency between the baseline model and MaskDM on data of varying resolutions. The baseline model employs exact same training settings as the fine-tuning stage of MaskDM.

## 4.3 IMAGE SYNTHESIS

To thoroughly evaluate and demonstrates the efficacy of our proposed training paradigm, we conduct multiple experiments on image synthesis tasks with different resolutions, including the CelebA $64 \times 64$, CelebA $128 \times 128$, and CelebA-HQ $256 \times 256$. Due to the scattered evaluation results reported in previous works, we have to compare the MaskDM models generated by our framework with models produced by various methods, as shown in Tab. 2 to 4.

Specifically, on CelebA $64 \times 64$, we adopt MaskDM-S to compare with three other models with comparable parameter sizes. In particular, our implemented MaskDM-S is most similar to U-ViT-small Bao et al. (2022)(Tab. 2) , with the only difference that we remove the final convolutional layer. After loading the best pre-trained weight reported in Tab.1b, we fine-tune MaskDM-S for 350k steps on 2 V100 GPUs and achieve an FID score of 2.2. The overall training takes approximately 2.09 V100 days. Our result significantly surpasses the FID scores reported by other works shown in Tab.2.

We find few research efforts on employing diffusion models with pure ViT-based architecture for dataset of resolution larger than $128 \times 128$. This is partially attributed to the training challenges caused by the lack of inductive bias in ViT and the computation complexity associated with the attention mechanism (Peebles & Xie, 2022; Bao et al., 2022). On CelebA $128 \times 128$, to the best of our knowledge, GenViT Yang et al. (2022) is the solitary ViT-based diffusion model for this specific task, although demonstrating limited generation quality. As such, we train a baseline diffusion model for 550k steps with identical settings (architecture, fine-tuning hyper-parameters and computational cost) as our MaskDM-B model using the objective detailed in Eq. 1 for comparison. As previously mentioned in Sec. 4.2, we find that manually adjusting the mask rates and training steps during the entire pre-training stage leads to better model performance. Specifically, we pre-train a MaskDM-B with a 70% mask rate for the beginning 50k steps and with a 30% mask rate for the remaining 350k

steps. Then we fine-tune the model for 100k steps and achieve the lowest FID score of 6.83 among all compared models. The training takes approximately 3.96 A100 days on 2 A100 GPUs, comprised of 3.26 A100 days pre-training and 0.69 A100 days fine-tuning.

On CelebA-HQ $256 \times 256$, we also manually adjust mask rates and pre-train a MaskDM-B with a 90% mask rate for the beginning 200k steps and with a 50% mask rate for the remaining 500k steps. Subsequently, we fine-tune the model for 100k steps and achieve an FID score of 6.27. Training this MaskDM-B model takes 12.19 A100 days on 2 A100 GPUs, which comprises of 9.86 A100 days for pre-training and 2.33 A100 days for fine-tuning. We build a baseline model as previously done for CelebA $128 \times 128$, and only achieves an FID score of 24.83 at the training cost of 18.28 A100 days. Comparing with other methods that either optimize models with adversarial training (Esser et al., 2021; Karras et al., 2017; Xiao et al., 2021) or training based on UNet in raw pixel space (Song et al., 2020b; Kim et al., 2021; Choi et al., 2022), our MaskDM-B model achieves the lowest FID score utilizing U-ViT architecture with comparable model parameters. There are also studies (Vahdat et al., 2021; Rombach et al., 2022) on improving training efficiency for diffusion models by focusing on the latent space. In comparison with these last two methods, our MaskDM-B model significantly outperforms LSGM Vahdat et al. (2021), but obtains worse performance than LDM Rombach et al. (2022). It is important to note that LDM Rombach et al. (2022) utilizes an extra VAE Kingma & Welling (2013) as the feature extraction model, which is pre-trained on ImageNet $256 \times 256$ for hundreds of thousands of steps. In contrast, we only train one 102M-parameter ViT-based model consistently on CelebA-HQ $256 \times 256$ without any extra data. We anticipate our performance could be further enhanced by incorporating advanced training techniques (Karras et al., 2022; Dhariwal & Nichol, 2021).

We further present comparisons of the training efficiency between baseline and MaskDM models on these datasets in Fig. 4. For reaching similar FID scores (shown in vertical axes), MaskDM models can save 60% ($\sim$5/12 in Fig. 4(a)) to 80% ($\sim$30/165 in Fig. 4(c)) training hours than DDPM models. Moreover, the reduction in training time increases as the data dimensionality increases. We further select pairs of example human face images (Fig. 4(d)) to provide a qualitative comparison between the synthesis results obtained by pre-trained and baseline model for reaching the same FID scores. The more realistic synthesis images sampled from MaskDM evidently indicate better generation quality. We note that the training curve is saturated for several pre-training steps. In practice, however, we find more pre-training time eventually leads to better FID scores, consistent with findings in Sec.4.2.

## 4.4 GENERALIZABILITY OF MASKED PRE-TRAINING

As previously mentioned in Section 1, we expect a MaskDM model to have desirable generalizability to facilitate fine-tuning for downstream tasks. In the following experiments, we assess the generalizability of MaskDM models by fine-tuning them with different datasets and training paradigms (Fig. 5a), and pre-training them with limited data (Fig. 5b). Training details are presented in Appendix 9.

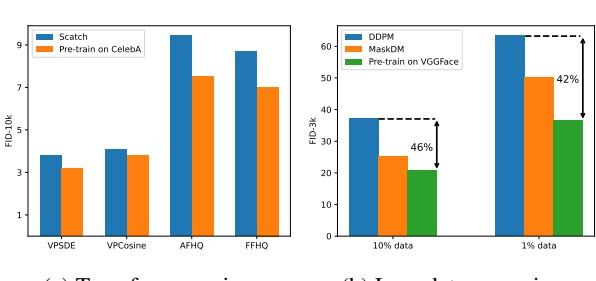

(a) Transfer scenarios      (b) Low-data scenarios

We first choose CelebA $64 \times 64$ as the source pre-training dataset and consider the scenario where the diffusion model training paradigms are not aligned across pre-training and fine-tuning (left bars in Fig. 5a). Specifically, we utilize DDPM in masked pre-training and adopt VPSDE Song et al. (2020b) or VPCosine Nichol & Dhariwal (2021) in fine-tuning. We also collect the pre-trained models from Sec.4.2 and fine-tune them on another two different datasets, i.e., FFHQ Karras et al. (2019) and AFHQ Choi et al. (2020), with all images resized to the resolution of $64 \times 64$. This creates data distribution shifts between the pre-training and fine-tuning datasets. As shown in Fig. 5a), when compared with models trained from scratch for each setting, pre-trained models demonstrate clear stronger generalizability for both training paradigm and data distribution shifts.

Then we construct two small training datasets, one with 3000 images(10%) and the other with 300 images(1%), from the CelebA-HQ $256 \times 256$ dataset. For each dataset, we maintain similar

computational expenses for training models from scratch for 200k steps and taking 200k-step pre-training with a mask rate of 50% followed by 50k-step fine-tuning. As illustrated in Fig. 5b), the final FID scores evaluated on the complete CelebA-HQ $256 \times 256$ dataset demonstrate that our proposed training framework significantly improve the quality of generated images. Moreover, we leverage another model, which is pre-trained on the VGGFace2 $256 \times 256$ dataset (containing approximately 3M training images) for 200k steps with a mask rate of 90% for comparison. Fig. 5b) shows that this leads to further improvements in the generation performance, underscoring the potential for tackling synthesis tasks facing with data scarcity by integrating a diffusion model that is roughly pre-trained on a large, analogous and in-house dataset.

## 5 RELATED WORK

**Efficient training for diffusion models** has drawn significant attention in generative model litera-ture (Chang et al., 2023; Esser et al., 2021; Chang et al., 2022). Since the considerable training costs associated with diffusion models have presented as an obstacle that impedes the advancement of the research community, there is an urgent need for the application of these methods in the realm of diffusion models. To tackle this challenge, two recent works (Rombach et al., 2022; Vahdat et al., 2021) propose latent diffusion models (LDM). These models are trained on low-dimensional features, extracted by a pre-trained VAE Kingma & Welling (2013), to alleviate the computation burden for approximate extraneous details inherent in high-dimensional data. Based on LDM, MDT Gao et al. (2023) and MaskDiT Zheng et al. (2023) are proposed to incorporate masking into the latent space during training and achieve further improvement on the training efficiency. Note that these training paradigms need to be tailored for specific tasks, while our pre-trained MaskDM can be generalized to various image synthesis tasks. Besides, with optimized mask configurations, we show that models pre-trained with our framework also maintain strong generation quality. Another line of works aim at reducing training overhead in the raw pixel space, either by re-weighting the denoising objective (Choi et al., 2022; Hoogeboom et al., 2023; Hang et al., 2023) or refining the hyper-parameters of diffusion process (Kim et al., 2021; Wu et al., 2023), which is parallel to and compatible with our approach. Additionally, Wang et al. (2023) proposes to train diffusion models alternately on both images and cropped patches. This method can be seen as a specialized instance of our method, given their stochastic masking schedule is modified as a progressively adapted masking and replace their U-Net Ronneberger et al. (2015) with U-ViT. In comparison, our proposed two-stage framework achieves significant training acceleration, especially on high-dimensional image data.

**ViT-based diffusion models** have been studied in several recent researches (Bao et al., 2022; Peebles & Xie, 2022; Cao et al., 2022; Yang et al., 2022), which take vision transformer(ViT) as the backbone for building diffusion models. These studies have analyzed the factors that affect the quality of generated samples and empirically find that models with smaller patch size tend to produce better results. Consequently, it aggravates the heavy computational burden associated with adopting ViT in diffusion models, including the high CUDA memory usage and lengthy training times. Instead of focusing on refining the architecture, we addresses this challenge by reducing the training time spent on the cumbersome approximation of full-resolution data, therefore reducing the overall computational expenses for training ViT-based diffusion models.

## 6 CONCLUSION

In this work, a masked pre-training approach is proposed to improve the training efficiency for diffusion models in the context of image synthesis. We design a masked denoising score matching objective to guide the model for learning a primer distribution that shares some diverse and important features, conveyed in group of marginals, with the target data distribution. We empirically investigate various masking configurations for their impacts on model performance and training efficiency, and evaluate our approach using U-ViT for image synthesis in the pixel space on several different datasets. The evaluation results show that our approach substantially reduces the training cost while maintaining high generalization quality, outperforming the standard DDPM training method by a significant margin. We also conduct experiments for evaluating the generalizability of models pre-trained through our approach in the cases of training paradigm mismatch, data distribution shift, and limited training data. We demonstrate that our approach suffices to produce a pre-trained model with strong generalization capabilities.

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

## A  MODEL CONFIGURATIONS

| Model | Patch size | Depth | Dim | MLP Dim | Attn Heads | Params |
|---|---|---|---|---|---|---|
| MaskDM-S | 4 | 13 | 512 | 2048 | 8 | 44M |
| MaskDM-B | 4 | 12 | 768 | 3172 | 12 | 102M |

Table 5: Details of MaskDM models

## B  IMPLEMENTATION DETAILS

| Dataset | CelebA | | | | CelebA | CelebA 128 × 128 | CelebA 256 × 256 |
|---|---|---|---|---|---|---|---|
| Experiment | Tab.1 | | | | Tab.2 | Tab.3 | Tab.4 |
| **pre-train** | | | | | | | |
| Masking | any | | | - | 2x2 block-wise | 4x4 block-wise | 4x4 block-wise |
| Mask rate | 10% | 50% | 90% | - | 50% | 70%, 50% | 90%, 50% |
| Lr | 1e-4 | 2e-4 | 2e-4 | - | 2e-4 | 2e-4, 1e-4 | 2e-4 |
| Batch size | 128 | 256 | 512 | - | 256 | 256, 128 | 128, 64 |
| Steps | 50k | any | 50k | - | 150k | 50k, 350k | 200k, 500k |
| Gradient clip | - | | | | - | 1.0, 1.0 | -, 1.0 |
| Warmup | - | | | | 5k steps | 5k steps, 5k steps | -, 5k steps |
| Noise schedule | Linear | | | | Linear | Cosine | Cosine |
| **fine-tune** | | | | | | | |
| Lr | 1e-4 | 1e-4 | 1e-4 | 1e-4 | 2e-4 | 5e-5 | 1e-5 |
| Batch size | 128 | 128 | 128 | 128 | 128 | 64 | 32 |
| Steps | 200k | 200k | 200k | 250k | 350k | 100k | 100k |
| EMA setting | 0.999 update every 1 | | | | 0.9999 update every 1 | 0.999 update every 1 | 0.999 update every 1 |
| Gradient clip | - | | | | - | 1.0 | 1.0 |
| Warmup | - | | | | 5k steps | 5k steps | 5k steps |
| **shared parameters** | | | | | | | |
| Model | MaskDM-S | | | | MaskDM-S | MaskDM-B | MaskDM-B |
| Noise schedule | Linear | | | | VPSDE | Cosine | Cosine |
| Horizontal flip | - | | | | 0.5 | - | - |
| **sampling** | | | | | | | |
| Sampler | DDIM | | | | Euler-Maruyama | DDIM | DDIM |
| Sampling steps | 500 steps | | | | 1000 steps | 500 steps | 500 steps |
| Num of samples | 10k | | | | 50k | 50k | 50k |

Table 6: Hyper-parameters of experiments in Tables

**Lower LR used in 10% masked pre-training in Tab.1.** In early experiments, we observe that the model yields a poor performance when the learning rate is set to 2e-4, using 128 batch size. Therefore, we scale the learning rate linearly according to the batch size and use 1e-4 in our experiments.

## C  UNSTABLE PRE-TRAINING UNDER 90% MASK RATE

In early experiments, we consistently observe that the model hardly converges when trained at 90% mask ratio. Therefore, we investigate the impact of various factors on the pre-training process, including batch size, learning rate, noise schedule, gradient clipping, and learning rate schedule (Warmup). As shown in Fig.6, increasing the batch size from 128 to 2048 still results in divergence. Besides, when we reduce the learning rate from 2e-4 to 1e-4 and the batch size from 256 to 128, we observe a stable and gradual plateau in the FID score of the model after 200k training steps.

| Dataset | CelebA | CelebA $128 \times 128$ | CelebA $256 \times 256$ |
|---|---|---|---|
| **pre-train** | | | |
| Masking | 2x2 block-wise | 4x4 block-wise | 4x4 block-wise |
| Mask rate | 50% | 70%, 50% | 90%, 50% |
| Lr | 2e-4 | 2e-4, 1e-4 | 2e-4 |
| Batch size | 256 | 256, 128 | 128, 64 |
| Steps | 50k | 50k, 350k | 200k, 500k |
| Gradient clip | - | 1.0, 1.0 | -, 1.0 |
| Warmup | - | 5k steps, 5k steps | -, 5k steps |
| Noise schedule | Linear | Cosine | Cosine |
| **fine-tune** | | | |
| Lr | 1e-4 | 5e-5 | 1e-5 |
| Batch size | 128 | 64 | 32 |
| Steps | 200k | 100k | 100k |
| EMA setting | 0.9999 update every 1 | 0.999 update every 1 | 0.999 update every 1 |
| Gradient clip | - | 1.0 | 1.0 |
| Warmup | - | 5k steps | 5k steps |
| **shared parameters** | | | |
| Model | MaskDM-S | MaskDM-B | MaskDM-B |
| Noise schedule | Linear | Cosine | Cosine |
| Horizontal flip | - | - | - |
| **sampling** | | | |
| Sampler | DDIM | DDIM | DDIM |
| Sampling steps | 500 steps | 250 steps | 250 steps |
| Num of samples | 10k | 10k | 10k |

Table 7: Hyper-parameters of experiments in Fig.4. For the baseline model, we use exactly the same hyper-parameter settings as in the fine-tuning of MaskDM. In addition, the training step is set to 250k, 550k and 800k steps for baseline model on CelebA, CelebA $128 \times 128$ and CelebA-HQ$256 \times 256$, respectively, maintaining a consistent computational cost with the MaskDM counterpart.

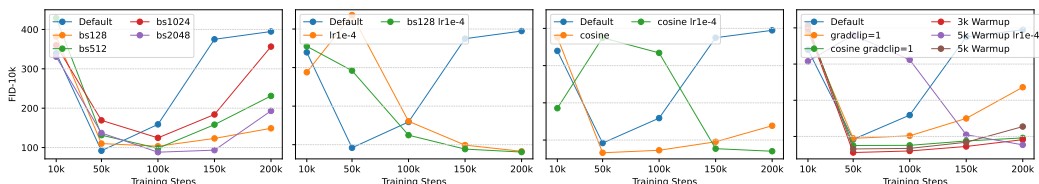

Figure 6: Unstable training investigation on CelebA dataset. We maintain a fixed mask ratio of 90% and adopt the parameters used in pre-training experiments at the 50% mask ratio (See Tab.6) as default setting. When employing Warmup, the learning rate follows a linear schedule starting from 5e-5.

However, this more conservative hyperparameter setting leads to a relatively slower convergence speed, necessitating a longer training time to achieve a similar level of performance for the model.

Subsequently, we adopt a cosine schedule in place of the linear schedule, and the resulting FID score curve demonstrates the superiority of the cosine schedule in improving the training stability of the model. It effectively mitigates the convergence issues observed with the linear schedule. Additionally, we implement extra optimization strategies, such as Warmup and gradient clipping, which also contribute to a more stable convergence of the model.

| Dataset | CelebA | FFHQ $64 \times 64$ | AFHQ $64 \times 64$ |
|---|---|---|---|
| **pre-train** | | | |
| Masking | | 2x2 block-wise | |
| Mask rate | | 50% | |
| Lr | | 2e-4 | |
| Batch size | | 256 | |
| Steps | | 50k | |
| Noise schedule | | Linear | |
| **fine-tune** | | | |
| Lr | | 1e-4 | |
| Batch size | | 128 | |
| Steps | | 200k | |
| EMA setting | | 0.999 update every | |
| **shared parameters** | | | |
| Model | | MaskDM-S | |
| Noise schedule | VPSDE, VPCosine | Linear | Linear |
| Horizontal flip | 0.5 | - | - |
| **sampling** | | | |
| Sampler | DPM-Solver | DDIM | DDIM |
| Sampling steps | 50 steps | 500 steps | 500 steps |
| Num of samples | 10k | 10k | 10k |

Table 8: Hyper-parameters of experiments in Fig.5a. Following the parameters used during fine-tuning MaskDM, the baseline models are trained for 250k steps. We employ DPM-Solver Lu et al. (2022) to generate samples on CelebA dataset.

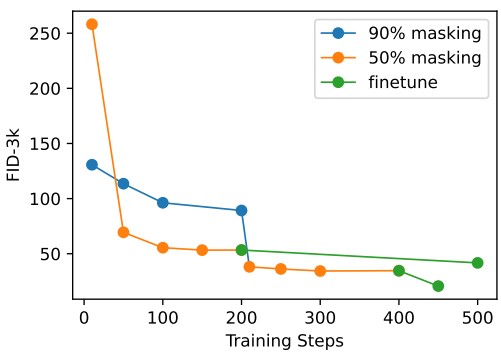

Figure 7: The masked pre-training process.

## D    THE CONVERGENCE SPEED OF PRE-TRAINING

As illustrated in Fig.7, the MaskDM model converges rapidly during pre-training. Besides, the progressive mask rate schedule demonstrates a better performance.

## E    ADDITIONAL VISUALIZATION RESULTS

| Dataset | CelebA-HQ $256 \times 256$ 10% or 1% | VGGFace2 $256 \times 256$ |
|---|---|---|
| **pre-train** | | |
| Masking | 4x4 block-wise | 4x4 block-wise |
| Mask rate | 50% | 90% |
| Lr | 2e-4 | 2e-4 |
| Batch size | 64 | 256 |
| Steps | 200k | 200k |
| Noise schedule | Cosine | Cosine |
| **fine-tune** | | |
| Lr | 5e-4 | |
| Batch size | 64 | |
| Steps | 50k | |
| EMA setting | 0.999 update every 1 | |
| **shared parameters** | | |
| Model | MaskDM-S | |
| Noise schedule | Cosine | |
| Horizontal flip | 0.5 | |
| **sampling** | | |
| Sampler | DDIM | |
| Sampling steps | 250 steps | |
| Num of samples | 3k | |

Table 9: Hyper-parameters of experiments in Fig.5b. Following the parameters used when fine-tuning MaskDM, the baseline models are trained for 200k steps.

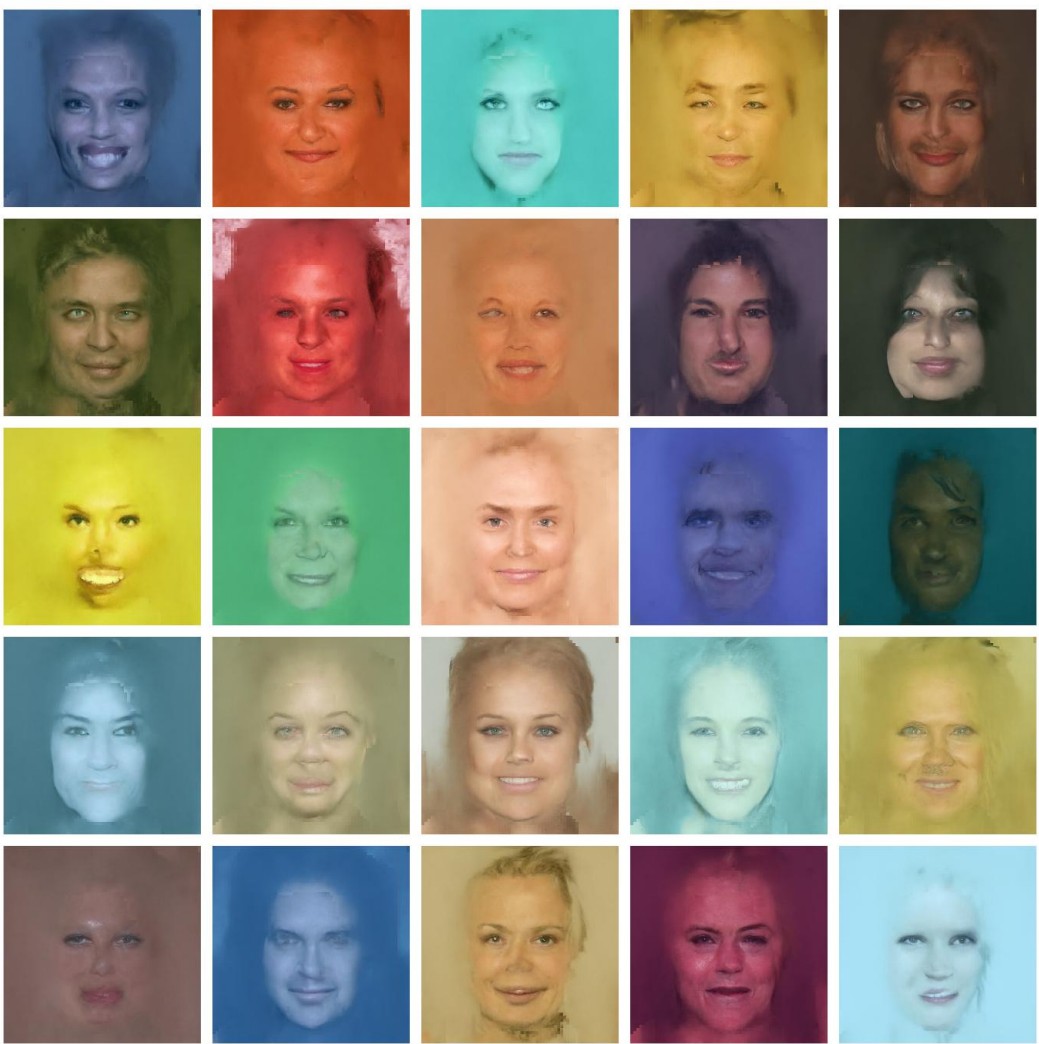

Figure 8: Samples generated by a pre-trained MaskDM model given a masking strategy of cropping and 90% mask rate.

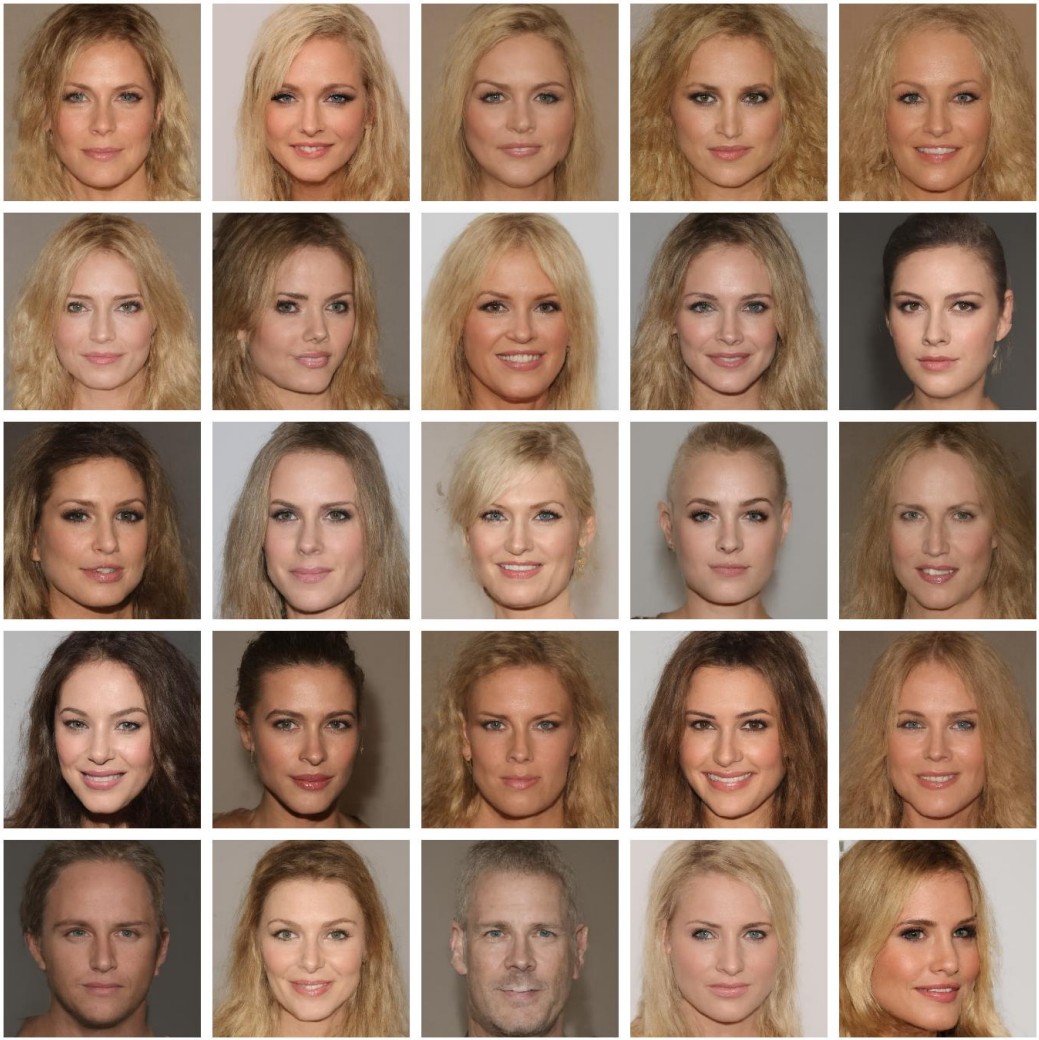

Figure 9: Uncurated samples generated by a pre-trained MaskDM model (4x4 block-wise masking and 90% mask rate).

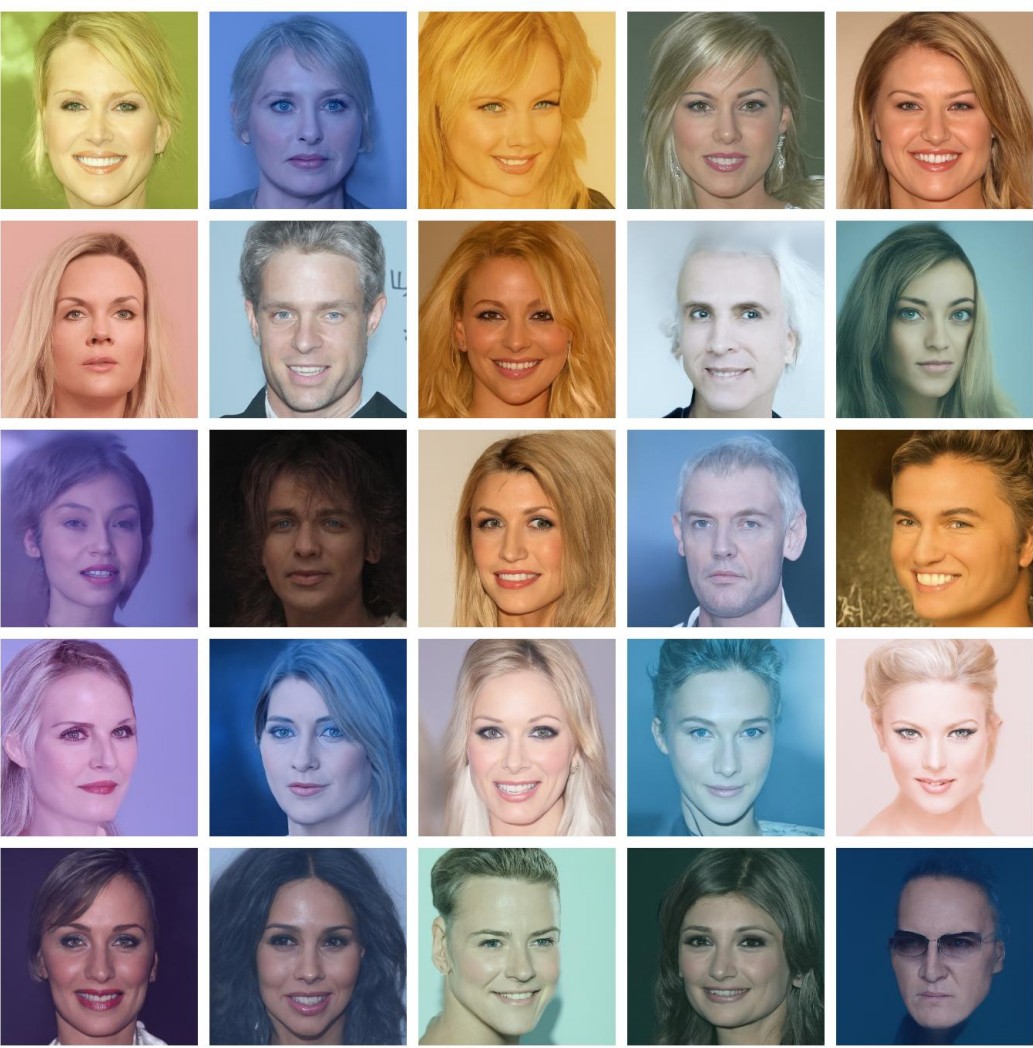

Figure 10: Uncurated samples generated by a pre-trained MaskDM model (4x4 block-wise masking and 50% mask rate).

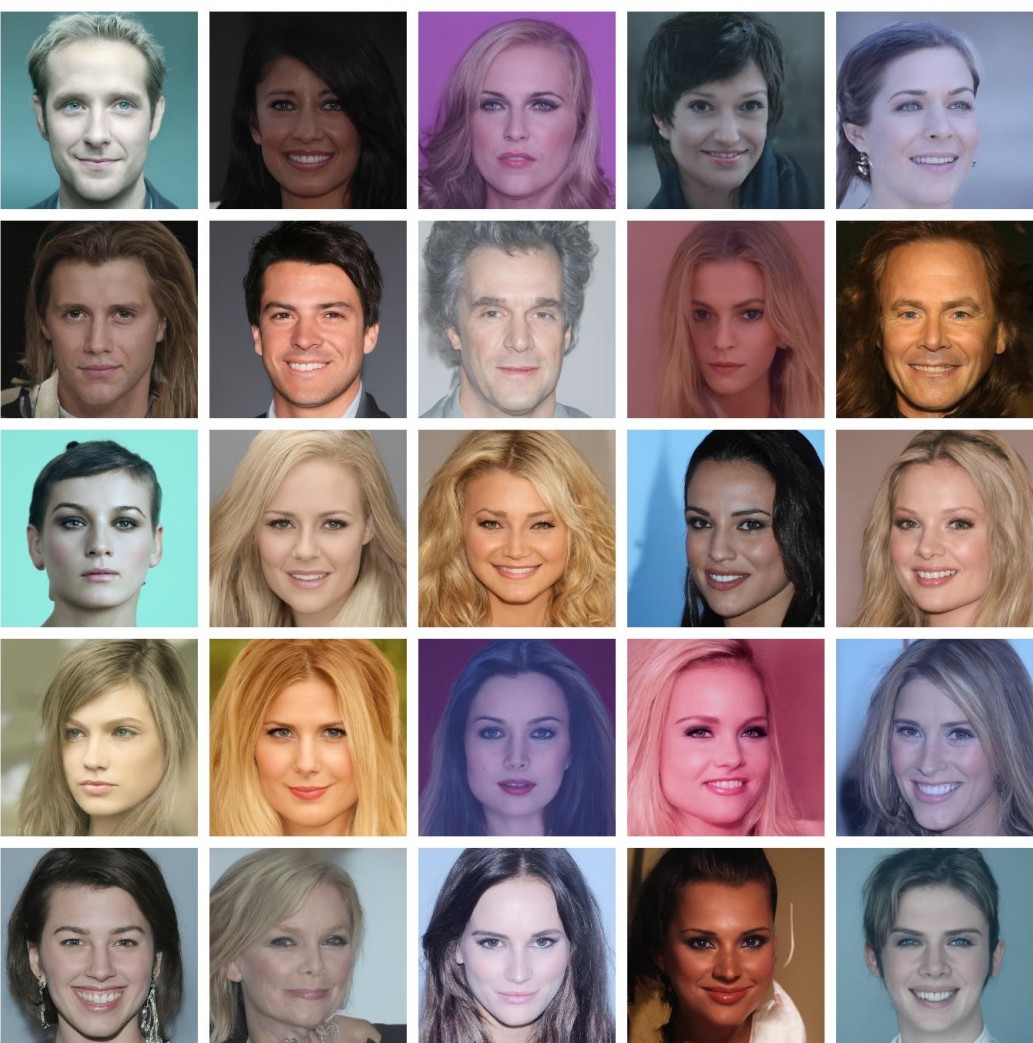

Figure 11: Uncurated samples generated by a pre-trained MaskDM model, given the configuration of 4x4 block-wise masking and 50% mask rate, after loading weights that are pre-trained at 90% mask rate. The training takes 100k steps.

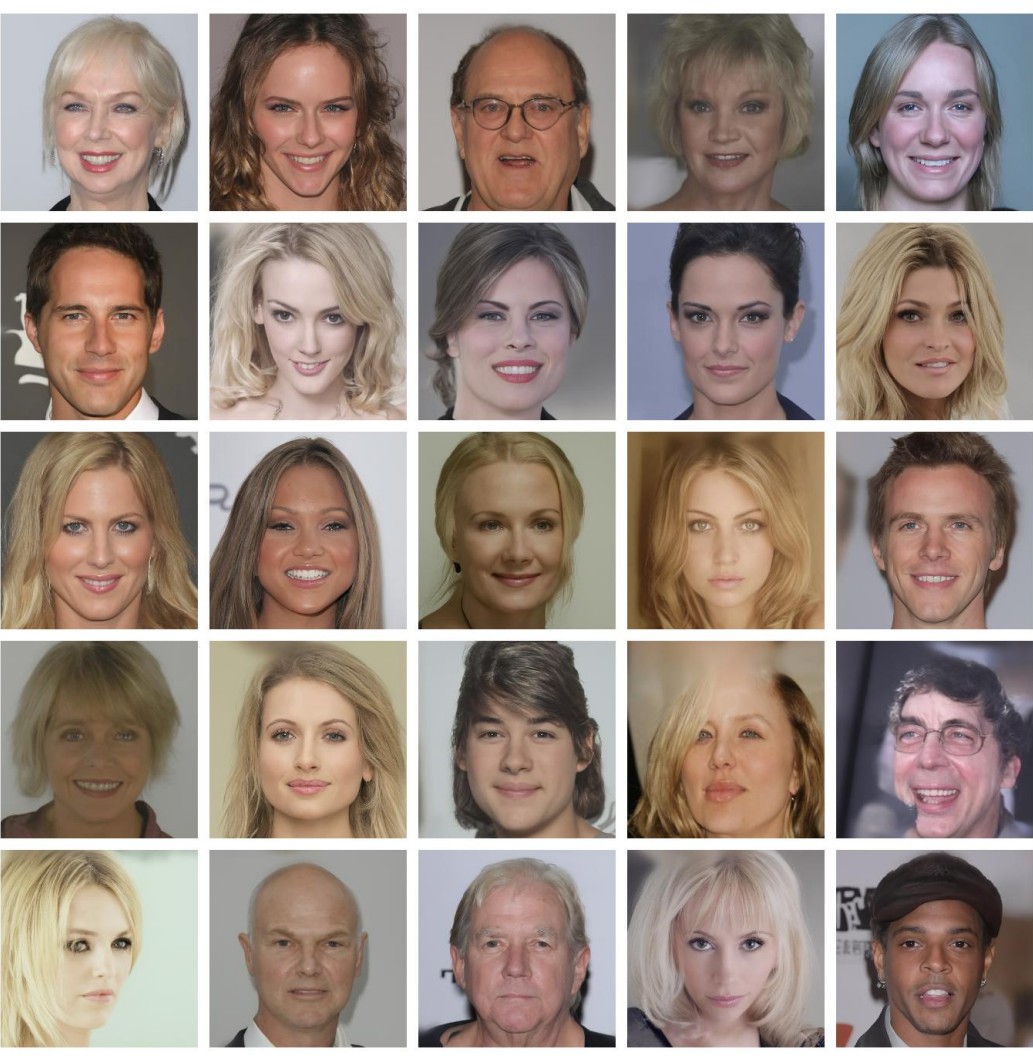

Figure 12: Uncurated samples generated by a pre-trained MaskDM model, given the configuration of 4x4 block-wise masking and 50% mask rate, after loading weights that are pre-trained at 90% mask rate. The training takes 500k steps.

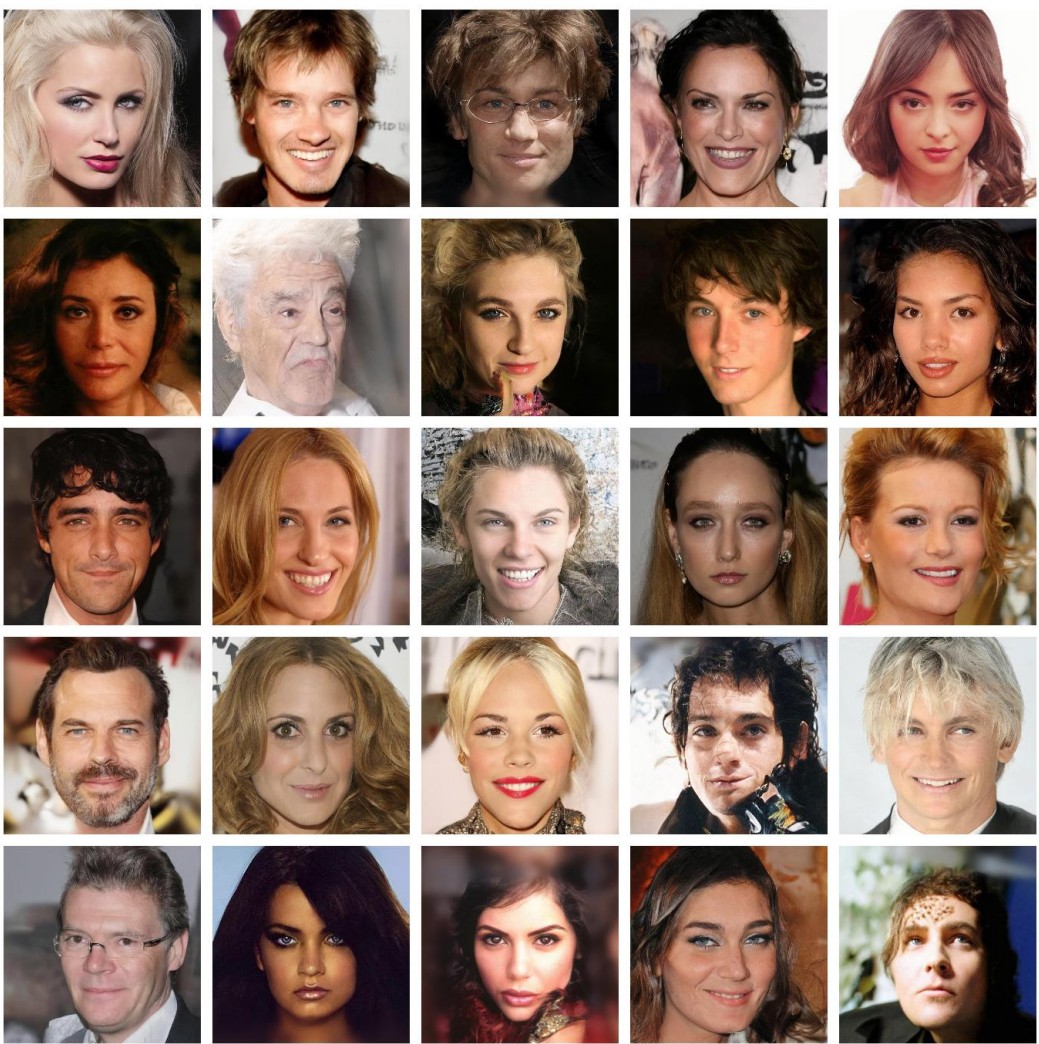

Figure 13: Uncurated samples generated by our MaskDM-B model trained on the CelebA-HQ $256 \times 256$ dataset in Tab.4.

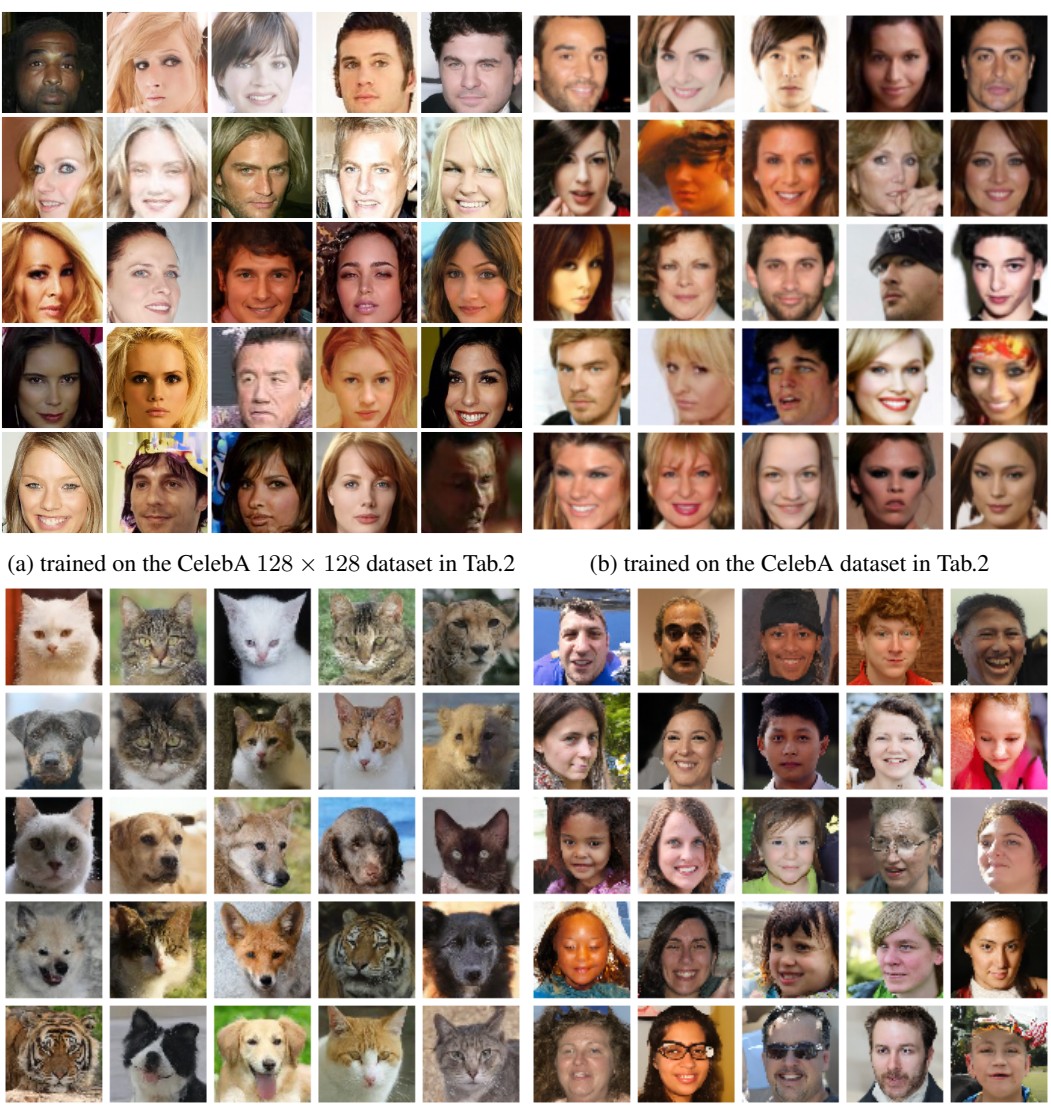

(a) trained on the CelebA $128 \times 128$ dataset in Tab.2

(b) trained on the CelebA dataset in Tab.2

(c) fine-tuned on AFHQ $64 \times 64$ in Fig.5a.

(d) fine-tuned on FFHQ $64 \times 64$ in Fig.5b.

Figure 14: Uncurated samples generated by our MaskDM-S models.

