# OpenReview forum: "Masked Diffusion Models are Fast Distribution Learners"
_ICLR.cc/2024/Conference — ICLR 2024 Conference Withdrawn Submission_

### Official Review · Reviewer_SYPq · 2023-10-28

**Soundness:** 2 fair
**Presentation:** 2 fair
**Contribution:** 2 fair
**Rating:** 5
**Confidence:** 4

**Summary:**

This work design a two-stage training framework for improving diffusion model training efficiency. They emphasized that the proposed pre-training stage is a plug-and-play stradegy, which can be integrated with existing diffusion models.

**Strengths:**

1.	The first work to employ masked image pre-training into diffusion models to accelerate the convergence and save the training costs.
2.	Extensive experiments are conducted and great performance is achieved.

**Weaknesses:**

1.	Masked image pre-training is a popular stragedy to provide the ViT with image prior. Observed from Algorithm 1, there is few novelty compared to such kind of works.
2.	Though authors present 2D Swiss roll as an intuitive example, theoretical support for the proposed method of approximating the primer distribution should be improved.
3.	Experiments on deploying the proposed pre-trained method onto more existing diffusion methods are desired to prove the characteristics of plug-and-play.
4.	Please clarify the fairness of comparisons with other diffusion-based methods, e.g., backbone.

**Questions:**

Please see weaknesses

---

### Official Review · Reviewer_DGdo · 2023-10-31

**Soundness:** 2 fair
**Presentation:** 3 good
**Contribution:** 2 fair
**Rating:** 5
**Confidence:** 4

**Summary:**

This paper proposed a two-stage training framework named Masked Diffusion Models (MaskDM) to reduce the training overhead of diffusion models. In the first stage, the authors devised various masking strategies to build a pre-trained model aimed at learning the primer distribution of target data distribution. In the second stage, the pre-trained model can be fine-tuned for specific generation tasks efficiently. Experimental results show that the proposed framework outperforms comparative methods in image generation performance and generalization performance.

**Strengths:**

1. The proposed Masked Diffusion Models (MaskDM) to reduce the training overhead of diffusion models is intuitive, and the analysis from the perspective of primer distribution is interesting.
2. The proposed two-stage training framework allows MaskDM to generalize to a new dataset with only a small amount of data for fine-tuning.
3. The paper is well-written and easy to follow.

**Weaknesses:**

1. The authors state that “our masked pre-training technique can be universally applied to various diffusion models that directly generate images in the pixel space, aiding in the learning of pre-trained models with superior generalizability.” The proposed Masked Diffusion Models are well suited to use VIT as the backbone. However, the current mainstream diffusion models use CNNs as the backbone. There is still insufficient evidence to determine whether Transformers contribute to the performance of diffusion-based image generation tasks. Therefore, this article lacks experimental or theoretical analysis on combining the proposed framework with previous CNN-based Diffusion models.

2. In Figure 4, the authors discussed the training efficiency of MaskDM compared to baseline DDPM. However, this baseline DDPM is implemented based on VIT. The reviewer believes it is also necessary to compare it with the CNN-based DDPM model to illustrate the training efficiency of the proposed model.

3. Some strategies to reduce the training overhead of diffusion models, such as Latent Diffusion Models and Cascaded Diffusion Models [1] are similar to the proposed method. The authors did not discuss the differences between these methods and the advantages of the proposed method in this article.

    * [1] Ho, Jonathan, et al. "Cascaded diffusion models for high fidelity image generation." The Journal of Machine Learning Research 23.1 (2022): 2249-2281.

4. Similarly, the authors did not report the training efficiency of other strategies in Fig. 4, which is not enough to demonstrate the efficiency of the proposed model.

5. In Table 2, the FID of DDIM is 3.26. This FID corresponds to the “DDPM ($\eta=1$ and $\hat{\sigma}$)” situation of the original article of DDIM. Therefore, labeling it as DDIM here will cause misunderstanding.

**Questions:**

1. The proposed Masked Diffusion Model serves to reduce the training overhead of the diffusion model. What are the essential differences and advantages of the proposed model compared to Latent Diffusion Models and Cascaded Diffusion Models?
2. Figure 3 shows the images sampled by the pre-trained model under three different masking strategies. How did the author extend the model pretrained through masked denoising score matching to the full-image generation task?

---

### Official Review · Reviewer_GstE · 2023-11-03

**Soundness:** 2 fair
**Presentation:** 3 good
**Contribution:** 2 fair
**Rating:** 3
**Confidence:** 4

**Summary:**

This work proposes masking as a pretraining scheme for diffusion models. They claim that the proposed approach can learn the data distribution more effectively compared to previous work and improves the image generation quality in terms of FID. They evaluate their method on CelebA and CelebA-HQ.

**Strengths:**

1. The authors provide a thorough analysis with various ablation studies.

2. The problem of improving the distribution learning in the pretraining step of the diffusion models is interesting.

**Weaknesses:**

1. There are ambiguities in the manuscript. E.g. what is the point of learning a swiss roll distribution? This is mentioned as an example of learning the distribution, but it is not clear how it relates to the image generation task. Also, the paper's main claim in the title is on learning distributions, but there are no experiments supporting this claim. The paper only provides experiments on image generation quality based on FID and qualitative comparison. The distribution learning claim needs to be backed by performing more downstream tasks using the proposed model.

2. There has been a couple of works combining masked autoencoders and specifically ViTs with diffusion models. The authors only mention two of these, but do not contrast their methodology in detail against those, and do not compare their model's performance against them. Here are more papers proposing a similar idea. In general, this makes the novelty of the work limited.
[a] Wei, Chen, et al. "Diffusion Models as Masked Autoencoders." arXiv (2023).
[b] Pan, Zixuan, Jianxu Chen, and Yiyu Shi. "Masked Diffusion as Self-supervised Representation Learner." arXiv (2023).

3. The method is only evaluated using FID. There are much more metrics commonly used for image generation such as KID, IS, CLIP Score, Precision / Recall.

4. The performance gain is either marginal or worse e.g. compared to LDM.


Minor:

Fig. 5 on page viii does not have a caption.

The authors mention that they achieved a record value of 6.73 in FID. However, this value is never used in any table or text.

**Questions:**

1. What is the main benefit of the proposed method compared to [a]?

2. Could the proposed model be tested for other downstream tasks to evaluate the distribution learning performance?

---

### Official Review · Reviewer_mkao · 2023-11-05

**Soundness:** 3 good
**Presentation:** 3 good
**Contribution:** 2 fair
**Rating:** 3
**Confidence:** 4

**Summary:**

This paper proposes a masked pretraining mechanism for diffusion models to improve the training efficiency. By randomly masking the images, the training objective will be applied to these selected patches as opposed to the vanilla diffusion models that don't use this masking strategy. Experiments are conducted on CelebA and CelebA-HQ datasets showing that this strategy could speed up the training process.

**Strengths:**

1. To improve the training efficiency of diffusion models is an important problem.
2. The experimental results on CelebA-HQ show that the method performs better than the baseline.

**Weaknesses:**

1. The paper only shows results on CelebA and CelebA-HQ which is not sufficient. More results on different other datasets need to be present to further demonstrate the effectiveness of the proposed method.
2. Results on CelebA 64x64 and 128x128 from Figure 4 (a) and (b) did not show that the proposed method has significant advantages over the baseline.
3. In Table 4, the results for the baseline which is U-ViT are missing.

**Questions:**

In the related works part, the authors claim that the training paradigms of MDT and MaskDiT which are two similar works need to be tailored for specific tasks, could the authors clarify more on this? Both of them also use masked strategies to improve the efficiency of diffusion models.

---

### Official Review · Reviewer_QVNJ · 2023-11-06

**Soundness:** 3 good
**Presentation:** 2 fair
**Contribution:** 2 fair
**Rating:** 5
**Confidence:** 3

**Summary:**

This paper introduces a pre-training approach tailored for training diffusion models to generate images. It introduces a masked encoding algorithm and adapts the UNet architecture to transformers accordingly. These design choices are rigorously validated through a series of experiments conducted on CelebA/CelebA-HQ datasets, showcasing the superior performance of the proposed method compared to multiple baseline methods. Furthermore, the effectiveness of each individual submodule is empirically demonstrated.

**Strengths:**

- This work delves into a critical aspect of diffusion models, focusing on the enhancement of their training efficiency.

- The idea proposed in this work is notably straightforward, and the results it yields are indeed promising.

- The writing is generally clear and largely accessible.

- The inclusion of a comprehensive ablation study is a strong point, and the validation of design choices adds credibility to the approach.

**Weaknesses:**

**Inconsistent/missing baselines**: The paper presents varying baseline models in Tables 2-4, all of which are generative models that can be conceptually reused in these experiments. Alternating between baselines in different experiments can be perplexing and lead to potential misinterpretation.

**Vanilla U-ViT**: It is important to report the results of a vanilla U-ViT architecture without the masked pre-training, as this would help discern whether the performance improvement stems from the pre-training algorithm or the architectural modifications. What's even good is to include different transformer based diffusion model results. Note that I'm not demanding these experiments,  I just state this as a weakness of the work.

**Redundant/insufficient experiment**: The evaluation on both CelebA and CelebA-HQ datasets may yield similar insights, given their substantial similarity. In addition to these, it is customary for generative models to be evaluated on larger-scale datasets such as ImageNet and LSUN. These experiments can be down at lower resolutions (64x64) to manage computational costs effectively.

**Presentation**: IMHO, the story of the primer distribution and the marginal distribution, while potentially insightful, appears to complicate understanding. Given the straightforward nature of the work and its simple implementation, streamlining the presentation and perhaps relocating the swiss roll toy example to a separate section could enhance readability.

**Visual comparison**: A qualitative comparison between this work and other existing methods is notably absent, making it challenging to directly appreciate the advantages of this approach.

**Inconsistencies regarding VGGFace2**: The paper frequently references VGGFace2, particularly in the abstract, where it claims a 46% quality improvement through fine-tuning on just 10% data from a different dataset. However, detailed results and discussions related to VGGFace2 are conspicuously absent in the main paper. To substantiate these claims, it is crucial to provide pertinent results and discussions within the main body of the paper.

**Incomplete related work**:
- MAE for other vision tasks or even broader topics (nlp, robotics) should be mentioned.
- Pre-training methods for generative models (GANs, VAEs) should be mentioned as well. One example is "Contrastive Learning for Unpaired Image-to-Image Translation, Park et al., ECCV'20".
- MDT and MaskDiT are mentioned in related work but not compared to in experiments. What's the reason?

**Questions:**

**Redundancy in Equations**: Equations (1) and (2) appear to be identical, with the only difference being in the variable names. It's not clear why the same equation is rewritten with different variable names. Clarifying the purpose of this redundancy would be helpful.

**Choice of U-ViT Backbone**: The paper mentions the use of U-ViT as the backbone for efficiency. It would be beneficial to include the comparison between U-ViT to others and this can be down in a limited setting (smaller size). This can help readers understand the rationale behind the architectural decision.

**Pre-training time**: I must have missed this section, how long does it take for pre-training stage for 64x64 and 256x256 models?

**Details Of Ethics Concerns:**

This paper yields generative models for human faces. Even though it's trained on public dataset, there might be potential ethical concerns. I didn't check the license of CelebA/CelebA-HQ carefully, but just raise this to avoid any future trouble.

---

### Official Review · Reviewer_yJNq · 2023-11-08

**Soundness:** 3 good
**Presentation:** 2 fair
**Contribution:** 3 good
**Rating:** 5
**Confidence:** 3

**Summary:**

The authors propose a pre-training strategy that helps diffusion models train quicker and well as in an efficient manner. This technique, which is masking the input images is generalizable to transformer based backbones and across datasets. Overall they demonstrate that pre-training helps reduce the training time, as well as improves the FID score on generated images.

**Strengths:**

1. Proposes a simple to implement pre-training technique for improving diffusion model convergence and accuracy.
2. Show results on known dataset and compare against multiple other published models.
3. In theory section, they try to formulate why this works.

**Weaknesses:**

1. In experiments it doesn't detail the time advantage of pre-training.
2. In experiments no detail on amount of fine-tuning and ablations on sensitivity to it
3. Overall the masking seems very hyper parameter sensitive as detailed in 4.2. This is further detailed in Appendix

**Questions:**

1. How generalizable is it to general images, and not posed celeb-A data?
2. The concept of structure definition learnt during masking isn't explored in results. Directly relying on FID can be a weak measure of technique.